# Burnout and Belonging: How the Costs and Benefits of Youth Activism Affect Youth Health and Wellbeing

Jerusha Osberg Conner [1,*], Emily Greytak [2], Carly D. Evich [3] and Laura Wray-Lake [4]

1   Department of Education and Counseling, Villanova University, Villanova, PA 19085, USA
2   ACLU (American Civil Liberties Union), New York, NY 10004, USA
3   Human Development and Family Science, Purdue University, West Lafayette, IN 47907, USA
4   Department of Social Welfare, University of California, Los Angeles, CA 90095, USA
*   Correspondence: jerusha.conner@villanova.edu

**Abstract:** Engagement in youth activism has been linked to both positive and negative wellbeing. Drawing on survey results from a sample of 636 youth participants in the ACLU Advocacy Institute, this study finds that although youth generally report greater benefits from their activism than costs, the costs are significantly related to worse mental health, physical health, and flourishing, while benefits are positively associated with flourishing only. A sense of belonging to an activist community, however, emerges as a significant protective factor for mental health, physical health, and flourishing. Focus group respondents explain how peer support and a sense of belonging act as salves to burnout, the most common cost that youth activists in this sample report experiencing. They also identify three main sources of burnout: backlash in response to their efforts; pressure to be the savior generation; and the slow progress of change. This study advances understanding of the complex relationship between youth activism and wellbeing and raises implications for youth activists and those who support them.

**Keywords:** youth activism; health; wellbeing; burnout; belonging

## 1. Introduction

In recent years, health professionals have been sounding the alarm about a crisis in youth mental health and wellbeing [1]. From the COVID-19 pandemic to the murders of George Floyd, Breonna Taylor, and other Black individuals, and from catastrophic climate events to policies and rulings that have stripped away basic rights, Gen Z is coming of age in a precarious time. To channel the grief, rage, anxiety, or despair they feel about these various crises, some youth have turned to activism. Whether precipitated by national or local events (e.g., mass school shootings), institutional actions (e.g., an unsatisfactory response to incidents of racism on campus), personal experiences with violence and oppression, existential dread due to climate change, or some other array of factors, activism offers participants a conduit to greater feelings of control, empowerment, and hope as they work to confront and challenge injustice and threats to their survival. Researchers have found activist spaces to be powerful sites of radical healing, particularly for racially marginalized youth [2–4].

While it can generate positive health and wellbeing, activism can expose youth to greater mental and physical risks, including harassment, arrest, violence, and even death. Among some youth activists, a culture of guilt, or not sacrificing enough for the cause, can develop [5]. Spending more time on activism can come at the expense of other necessary functions, such as sleep, exercise, and schoolwork [6]. Burnout has emerged as a pressing concern among youth activists [7,8].

Therefore, when seeking to understand how youth activism impacts youth, it is important to consider both the benefits and the costs of engagement. To date, most of the research exploring how youth activism implicates health and wellbeing finds either positive

or negative impacts; however, some emerging research is starting to examine the "both, and" effects of youth activism on young people [4,5,9], and to consider the ways in which youth with different identities may experience these impacts differently (e.g., [10–12]). This study builds on that research by exploring the costs and benefits of activism among young people and examining how these costs and benefits relate to their wellbeing.

## 2. Youth Activism and Wellbeing

The research exploring the relationship between youth activism and wellbeing has conceptualized activism using a variety of frameworks, including critical consciousness, sociopolitical development, and empowerment theory. We define activism as taking action to effect change in an unjust status quo. As such, activism encompasses organizing as well as other actions undertaken to challenge inequities or injustice. Existing research has also examined different facets of wellbeing, including various indicators of mental health, risk behavior, and social-emotional functioning [13].

Overall, in their systematic review of 29 studies, Maker Castro and colleagues [13] concluded that research with adolescents tends to find positive relationships between activism and wellbeing; however, research with young adult samples, mostly college students, is much more mixed. Among these samples, most of the studies reviewed found either a negative effect or a mixed effect.

Results also vary by specific student identities, though not always in consistent ways. For example, Hope and colleagues [11] found that activism can serve a protective function for the mental health of Latine college students, but not for Black students in predominantly white institutions, while Ballard and colleagues [10] found that activism was associated with worse wellbeing outcomes for Latine, but not Black young adults. By contrast, Maker Castro and colleagues [13] concluded that the research on activism and wellbeing among adolescents of color coalesces around positive outcomes, including "better mental health and socioemotional health, positive youth development and fewer risk behaviors" (p. 6). Similarly, research has found that activism is associated with positive wellbeing for adolescents who identify as LGBTQ [14] and adolescents and young adults who identify as transgender [15], but not for queer college students [8]. Exploring the intersections of racial identities, sexual orientation, and gender identities, Kulick and colleagues [16] examined activism as a potential buffer against the negative impact of homophobic experiences among LGBTQ white college students and LGBTQ college students of color. Their findings underscored the complexity of the relationship, suggesting that activism did not ameliorate the impact of homophobia among the white students and exacerbated depression among the students of color. As Anyiwo and colleagues [17] point out, such "disparities in findings highlight a need for more research that examines the psychological impact of youth activism, with special consideration to how the unique aspects" (p. 88) of youths' identities shape both their activism and its implications for their development.

Indeed, Maker Castro and colleagues [13] concluded their literature review by calling for more research into the relationship between youth activism and wellbeing. They pointed to a need for work that examines activism in relation to physical health indicators; studies that explore mental health outcomes across diverse youth activists, such as across race and ethnicity, gender, and sexual identity; and research that adopts mixed methods approaches. This study responds to their call by focusing on the role of costs and benefits of activism, as these different experiences relate to youth activists' wellbeing.

## 3. Costs and Benefits of Engaging in Youth Activism

Numerous studies document the benefits associated with participation in youth organizing and activism. These benefits can be loosely grouped into educational or cognitive, civic, social-emotional, and professional realms.

Research has shown that involvement in organizing against educational inequities can shift some youth's orientation to schools, making them more committed to their education, even as they become more critical of the shortcomings of their schools [18–20]. One study

found that youth attending neighborhood schools significantly improved their grade-point-averages (GPAs) over the course of their involvement in organizing [21], and another study found that critical action during high school is associated with higher GPAs among Black and Latine students [22]. Compared to peers who were not engaged in activism, but who were involved in student government during high school, former youth organizers show higher college attendance aspirations as well as four-year college degree attainment [23], and some former youth organizers report that their selection of a major in college was heavily influenced by their formative experiences with activism [24,25]. In one national study, 57% of college student activists reported that their experiences as activists enhanced their academic performance [26].

Several studies report on what youth learn as a result of participating in activism. These learning outcomes include both civic knowledge, such as a greater understanding of how change happens and how various governmental agencies work, and civic skills, including communication skills [27–29], time management and planning skills [30], and leadership skills [31–34]. In addition, youth describe honing skills of critical social analysis through the political education they experience in activist spaces [27,35–42]. Activism may also impact racial identity development among racially marginalized adolescents, which may shape adolescents' social analysis and responses to discrimination [43]. Activism that challenges social injustice has also been associated with the development of civic efficacy or empowerment as well as greater critical reflection or analysis of inequalities [13,42].

Stronger interpersonal skills, along with enhanced social capital are frequently associated with participation in youth activism [44–47]. Because youth organizers enter organizing groups with relatively lower levels of social capital than matched peers who pursue other types of afterschool activities, the gains reported in the relational domain are particularly noteworthy [48]. Activism that challenges oppressive conditions can help young people develop collective problem-solving and leadership skills [49]. In addition, in the social-emotional realm, some research finds that youth report developing stronger emotional regulation and self-management skills as a result of their involvement in activism [48,50,51]. Activism provides opportunities for youth to learn to process, decenter, and harness emotions to facilitate their organizing efforts and support their own wellbeing in the process [52].

Beyond the educational, civic, and social-emotional benefits of youth activism, some youth activists may find new professional pathways. Studies have begun to illuminate how youth activism can shape the professional trajectories of former youth organizers as they enter adulthood [24]. One study found that former youth organizers credited their experiences in organizing with making them committed to empowering others through their careers, influencing their pursuit of socially oriented work opportunities [25]. Christens and colleagues [53] document how many of the former youth organizers they had studied a decade earlier have sustained their commitments to improving their city into adulthood. Taken together, this research suggests that youth activism can lead to consequential and lasting benefits for young people as well as their communities.

Despite this rich array of advantages, some work points to downsides of engaging in youth activism, particularly among older youth; however, there has been less research attention to the negative consequences of activism for young people than to the benefits. Most commonly, research has focused on the potential burnout that can result from activism. Burnout is characterized by stress that can be long-term and disruptive to wellbeing. Research documents the significant fatigue and burnout youth activists report [8,34,54–56]. Racial battle fatigue is a commonly experienced phenomenon among student activists of color, who must contend with relentless microaggressions and other forms of racism, including the expectation that they will serve to educate their white peers and white institutional agents about the pernicious effects of racism [7,57]. Other manifestations of burnout may be evident among youth with other identities; for example, white activists may experience white guilt and compassion fatigue [8]. Perhaps due to experiences of burnout or the significant time that activism takes, studies of college student activists have

found that activism can sometimes take a toll on students' academic performance [6,8]. Additionally, some work finds that student activists can suffer social consequences, such as the fraying of friendships and more strained relationships with family as a result of their activism [6,58]. With some types of activism, and for youth from racially minoritized groups, such as Black youth [59], activism can result in confrontations with police or counter-protesters, which can result in arrests, physical harm, and even death [60]. More research is needed to understand the costs of activism for young people.

## 4. The Current Study

This IRB approved study (UCLA 22-001642) extends the literature on youth activism by examining the costs and benefits of youth activism. Specifically, we address three central questions:

(1) How do youth activists weigh the costs and benefits of their activism? To what extent does the cost–benefit calculus differ by school level, racial and ethnic identity, gender identity, or sexual identity?

(2) How do the perceived costs and benefits of activism relate to youth activists' physical health, mental health, and flourishing? What other factors beyond costs and benefits are associated with wellbeing?

(3) How do youth activists describe the main costs of their activism and the factors that help sustain their engagement when experiencing these tolls?

By adding further evidence to the literature on costs and benefits of youth activism, this research can help advance understanding of the factors that shape the wellbeing of youth activists.

## 5. Methods

This study is based on data collected from participants in the 2021 ACLU National Advocacy Institute, a weeklong program for youth activists in their teens to early 20s held during the summer. The study uses participant data from the 2021 Institute, conducted virtually due to the COVID-19 pandemic. The Institute is designed to build participants' capacity for effective advocacy and organizing and to further their civic engagement and sociopolitical development. Programming contains a mix of keynote speakers, presentations, issue area electives, time to meet in affinity groups (spaces of shared identity), "homeroom" groups, community discussions, and social events.

The Institute is open to youth 15–23 years of age with an interest in civil liberties and social justice advocacy. No prior experience is required. Potential applicants are notified about the opportunity through several outreach methods: emails to ACLU members, posts on the national ACLU social media channels, and targeted paid advertisements on key social media channels (Instagram, TikTok) and search engines (e.g., Google). The 52 ACLU state/local affiliates also assist with outreach through emails and social media posts. In addition, Institute staff promote the opportunity to prospective youth and families at a small number of pre-college and camp fairs. Interested youth complete an online application and submit a letter of recommendation. There is a tuition fee for participants; scholarships for all who need them are provided.

### 5.1. Data Sources

To examine the potential factors related to the health and wellbeing of youth activists, this study used a mixed-methods design that included a questionnaire with both closed and open-ended items and follow-up focus groups. The mixed methods approach enabled us to pursue various mixing purposes, including complementarity, elaboration, and expansion because the focus groups were designed to explore findings that emerged in the quantitative data [55].

### 5.1.1. Questionnaire

We assessed participants' experiences as activists and organizers—including their current wellbeing—using a questionnaire administered online one week prior to the Institute. All 1046 registered participants received an email invitation to complete the voluntary questionnaire, and 636 completed it (60.8% response rate) (See Table 1). In addition to the items used for this study, the questionnaire included items relevant to assessing the impact of the Institute.

**Table 1.** Characteristics of Questionnaire Sample (N = 636).

| | |
|---|---|
| School Level (for Fall 2021) | High School: 78.6% (N = 496)<br>College: 16.5% (104)<br>Graduate School: 1.4% (9)<br>Not in School: 3.5% (22) |
| Gender (these are not mutually exclusive categories) | Female: 68.2% (N = 434)<br>Male: 18.7% (N = 119)<br>Nonbinary: 7.5% (N = 48)<br><br>Cisgender: 83.6% (N = 532)<br>Transgender: 4.2% (N = 27)<br>Genderqueer/Gender Fluid: 9.9% (N = 63)<br>Another gender identity: 0.01% (N = 9) |
| Race/ethnicity | Identifying as "person of color": 41.5% (N = 249)<br><br>(below are not mutually exclusive categories):<br>Black/African American: 11.2% (N = 71)<br>Native American/Indigenous/American Indian: 1.4% (N = 9)<br>Arab of Middle Eastern: 4.2% (N = 27)<br>Asian/South Asian: 20.0% (N = 127)<br>Latine: 12.3% (N = 78)<br>Native Hawaiian: 0.8% (N = 5)<br>White: 55.8 % (N = 374)<br>Bi/Multi Racial: 11.3% (N = 72)<br>Another race/ethnicity: 2.7% (N = 17) |
| Disability | 9.7% (N = 62) |
| Sexual identity (these are not mutually exclusive categories) | Gay/lesbian: 9.3% (N = 59)<br>Bisexual/pansexual: 30.3% (N = 193)<br>Heterosexual/straight: 40.3% (N = 256)<br>Queer: 14.8% (N = 94)<br>Asexual: 3.9% (N = 25)<br>Another sexual identity: 4.08% (N = 26) |
| Family income | $15,000: 8.2% (N = 18)<br>Between $15,000 and $29,999: 3.8% (N = 24)<br>Between $30,000 and $49,999: 5.2% (N = 33)<br>Between $50,000 and $74,999: 6.3% (N = 40)<br>Between $75,000 and $99,999: 7.7% (N = 49)<br>Between $100,000 and $150,000: 15.6% (N = 99)<br>Over $150,000: 27.2% (N = 173)<br>I have no idea: 23.3% (N = 148) |

**Independent variables**. Costs and benefits of activism were assessed through a 12-item, adapted version of the Political Costs Scale [61], and a 12-item, adapted version of the Political Benefits Scale [62] (See Appendix A). Items were rated on a seven-point scale from strongly disagree to strongly agree, and scales were summed, such that higher scores reflected higher costs and higher benefits, respectively. Both scales yielded strong alphas of 0.868 and 0.865, respectively (See Table 2).

**Table 2.** Descriptive and Reliability for Study Variables.

| Variables | Possible Range | Mean | SD | Cronbach's Alpha |
|---|---|---|---|---|
| Independent Variables—Health and Wellbeing | | | | |
| Physical Health | 1–5 | 3.74 | 0.97 | NA |
| Mental Health | 1–5 | 2.67 | 1.03 | NA |
| Flourishing Scale | 8–56 | 45.64 | 6.67 | 0.865 |
| Dependent Variables | | | | |
| Length of Time as Activist | 1–5 | 2.83 | 1.28 | NA |
| Perceived Effectiveness of Activism | 1–4 | 2.74 | 0.73 | NA |
| Number of Groups/Orgs Engaged With | 1–4 | 2.56 | 1.16 | NA |
| Belonging to Activist Community Scale | 3–15 | 10.70 | 2.69 | 0.842 |
| Activism Costs Scale | 12–84 | 42.24 | 14.00 | 0.868 |
| Activism Benefits Scale | 12–84 | 70.52 | 9.26 | 0.865 |

A range of other factors were included in the models as predictors of wellbeing. First, we assessed demographic factors. Participants self-reported the level of schooling in which they were currently enrolled, indicating high school, college, trade or vocational school, graduate school, or not enrolled as a student. Participants indicated their race and ethnicity by selecting all that applied from a list including Black/African American; American Indian, Native American, Alaska Native, Indigenous; Arab or Middle Eastern; Asian including South Asian; Latino/a/x; Native Hawaiian or other Pacific Islander; white or Caucasian; biracial or multiracial; and other race or ethnicity. For the purposes of this study, in order to have enough statistical power across groups, we combined the four race and ethnicity categories with the fewest respondents (American Indian, Native American, Alaska Native, Indigenous; Arab or Middle Eastern; Native Hawaiian or other Pacific Islander, and other race or ethnicity) into one category, unless participants selected two or more races or ethnicities, in which case we coded them as Bi- or Multiracial. Regarding gender, participants selected all that applied from female, male, genderqueer, gender fluid, non-binary, and two-spirited, and they also answered a separate question to indicate whether they identify as cisgender or transgender. Participants reported on sexual identity by selecting all that applied from lesbian or gay; bisexual or pansexual; queer; heterosexual/straight; asexual; or something else that they listed. We included transgender females, transgender males, and transgender nonbinary youth in the one overarching transgender category. We combined genderqueer, genderfluid, two-spirited, and nonbinary youth who were not transgender into one nonbinary category. Participants who selected anything other than "heterosexual/straight" were combined into the one "lesbian, gay, bisexual, or queer" (LGBQ) category.

Second, we assessed participants' experiences with activism and organizing through two items. One was length of time as an activist, with five response options including (1) I am just beginning with this institute, (2) less than a year, (3) 1–2 years, (4) 3–4 years, and (5) 5 or more years. The second item addressed the perceived effectiveness of their activism: youth responded on a four-point scale, (1) not at all effective, (2) not very effective, (3) somewhat effective, and (4) very effective.

Third, connectedness to activist communities was assessed through: (a) one item asking about the number of activist or organizing clubs, organizations, or groups in which participants are engaged on a scale of none, one, two, or three or more, and (b) the Belongingness to Activist Community Scale, a three-item measure adapted for this study from the Interpersonal Needs Questionnaire [62], with responses on a 5-point scale from (1) strongly disagree to (5) strongly agree on statements like, "I feel like I belong to an activist community". The alpha for the belongingness scale was 0.842.

**Dependent variables**. Three dependent variables pertained to wellbeing. One item each assessed participants' physical health and mental health, respectively (adapted from Bowling [63]). For these items, participants rated their general physical health and general

mental health on a 5-point scale including poor (1), fair (2), good (3), very good (4), and excellent (5). Flourishing was assessed with an eight-item Flourishing Scale, which describes "important aspects of human functioning ranging from positive relationships to feelings of competence to having meaning and purpose in life" [64], (p. 146). Participants responded on a seven-point scale from strongly disagree (1) to strongly agree (7) to items such as "I lead a purposeful and meaningful life". This scale yielded an alpha of 0.865. We analyzed these three wellbeing outcomes separately.

### 5.1.2. Focus Groups

After the surveys were collected, the research team conducted five one-hour focus groups (three for high school students; two for post-secondary youth). As part of their registration for electives, Institute participants were given the opportunity to volunteer to participate in focus groups on organizing and activism. Over 400 participants volunteered and 110 were selected randomly from the list of eligible youth. Those who returned consent forms were enrolled in the focus groups. If they did not complete the informed consent forms or indicated that they could not attend their assigned group, they were replaced by another randomly selected eligible volunteer. Parental consent for those under 18 years of age was obtained. Focus groups were conducted according to a focus group protocol developed in consultation with the National Political Advocacy Department (NPAD) Organizing Team at ACLU and the Advocacy Institute program staff. The one-hour protocol covered four topics: (1) Origin stories/trajectories, (2) Burnout and sustainability, (3) Organizing/advocacy identities and definitions, and (4) Future needs/resources. Each focus group was led by two members of the research team—one moderator and one note taker. Focus groups were recorded, and closed-captioned transcriptions were analyzed to identify key themes related to experiences of burnout and other perceived costs of activism.

### 5.2. Participants

Although we refer to the Institute participants in this study as youth "activists," we learned in our focus groups that not all participants feel comfortable applying the term activist to themselves, even though they engage in activist work. Some identified with other terms such as "advocate," "volunteer," or "organizer". For future survey administrations, we have added a question about how participants self-identify that will enable us to honor the ways in which participants see themselves and to explore whether these preferences reflect any differences in experiences.

Table 1 presents the demographic composition of the 636 survey respondents. The majority of respondents were high-school aged (78.6%), cisgender (83.6%), female (68.2%), white (55.8%), LGBQ (59.7%) youth, who were not living with a disability (90.3%), from upper middle class or upper-class families, with annual household incomes above $100,000 (42.8%).

A total of 28 young people (17 high school students and 11 post-secondary school youth) from 16 different states participated across the five focus groups. Participants ranged in age from 15–21 years (high school mean = 16.22 years, post-secondary mean = 20.38 years). The focus group respondents included 14 youth of color, ten who did not identify as youth of color, and four with unknown racial or ethnic identity. Allowing for biracial identification, participants included 12 who identified as white, seven as Asian/South Asian, five as Black, five as Latine, one as Indigenous/Native American, one as Native Hawaiian/Pacific Islander, and one as Arab/Middle Eastern. (Mindful that terminology continues to evolve, we have chosen to use the gender-neutral term Latine in this manuscript, rather than Latinx, in reference to our sample, because Latine is more consistent with the Spanish language than Latinx (McGhee, 2022). On the survey, however, we used the terminology "Latino/a/x".) Seventeen respondents identified as cisgender female, one as transgender female, three as cisgender male, and three as nonbinary or gender fluid. The gender identity of four respondents was unknown. In terms of sexual identity, 11 participants identified as

LGBQ and another 11 as straight or heterosexual. The sexual identity of six respondents was unknown.

*5.3. Analytic Approach*

We examined descriptive statistics on the variables of interest and conducted ANOVAs and *t*-tests to examine costs and benefits of activism across age, racial, gender, and sexual orientation groups. Stepwise hierarchical linear regressions were conducted to examine the role of costs and benefits in relation to wellbeing.

Focus group data were analyzed using a deductive and inductive coding schema. Analysis first proceeded deductively through a list of a priori codes derived from survey responses. In line with Zhang and Wildemuth's [65] qualitative content analysis process and following a close reading of all focus group transcripts, we reduced the data into manageable segments by identifying all sections of the transcripts in which participants discussed burnout or other costs associated with their activism. Next, we subjected these chunks of data to inductive coding, using a schema that was based on successive readings of these segments. The codes captured concepts that were not anticipated on the survey regarding the sources of burnout as well as the strategies respondents reported using for managing or mitigating it. Once all data were coded and discrepancies among coders resolved through discussion, we created matrix displays and used axial coding to identify further patterns.

To enhance the trustworthiness of our findings, we engaged a Youth Advisory Committee (YAC), a group of ten youth who had participated in the National Advocacy Institute during the year the data for this study were collected. The YAC was convened to provide guidance to researchers on formulating research questions, designing methods and protocols, and interpreting preliminary findings. Youth were compensated with gift cards following our meetings. At one meeting, we shared initial findings for this study with YAC members and asked for their feedback and advice about how to make meaning of the results. Their insights, especially about the importance of examining how youth with different identities interpreted the costs and benefits of activism differently, added another layer of richness to our inquiry. Their responses also affirmed that in studying burnout and belonging we were on an important track—one that could generate knowledge that would be relevant and useful to youth activists and their supporters. In addition, their final vetting of our work helped bolster the credibility of our claims.

## 6. Results

*6.1. Quantitative Findings*

Descriptive statistics showed that in this sample of youth activists, the benefits of activism considerably outnumbered the costs, with the benefits averaging 70.51 ($SD = 9.26$) and the costs averaging 42.24 ($SD = 14.00$). The most strongly endorsed benefit was a sense of identity: "my political and social views are an important part of who I am" (6.53, $SD = 0.85$). The second most endorsed benefit was "my activism has contributed to my personal growth" (6.42, $SD = 0.92$). On the costs side of the ledger, the most highly endorsed cost was activist guilt, specifically, "I have felt guilty for not doing enough for my activism (5.83, $SD = 1.46$), followed by "I have felt tired as a result of my activism" (4.71, $SD = 1.76$).

Comparisons of costs and benefits across identity groups showed that youth experienced these costs and benefits differently, depending on various dimensions of their identities. High school students reported significantly fewer costs than college students (HS $M = 41.14$, $SD = 13.96$; College $M = 45.55$, $SD = 13.32$, $F(3, 620) = 5.11$, $p = 0.002$), but the two groups did not differ in reported benefits and neither differed from youth enrolled in graduate coursework. Instead, youth who were not currently enrolled as students reported fewer benefits than college students (Not enrolled $M = 65.81$, $SD = 7.32$, College $M = 71.84$, $SD = 9.87$, $F(3, 617) = 2.63$, $p = 0.049$). Heterosexual or straight youth reported significantly fewer costs ($M = 36.3$, $SD = 12.79$) than LGBQ youth ($M = 47.00$, $SD = 13.40$, $t(581) = -9.60$,

$p < 0.001$), but they also reported fewer benefits ($M = 69.02$, $SD = 10.56$ vs. $M = 71.94$, $SD = 7.65$, $t(444) = -3.16$, $p < 0.001$). Both nonbinary youth and transgender youth reported more costs than cisgender male youth and cisgender female youth, $F(3, 585) = 16.23$, $p < 0.001$; however, nonbinary youth and transgender youth also reported more benefits than cisgender male youth, $F(3, 584) = 13.85$, $p < 0.001$ (though the two groups did not differ significantly from cisgender female youth in terms of benefits). Finally, there were no statistically significant differences in costs or benefits by race and ethnicity.

To assess the contributions of costs and benefits of activism to youth's wellbeing, we conducted three hierarchical regression models. Specifically, we regressed indicators of wellbeing (physical health, mental health, flourishing) on activism experience (perceived effectiveness of activism, length of time as an activist), connectedness (number of organizations/groups engaged with and belongingness to activist community), and costs and benefits, while controlling for racial identity, gender identity, and sexual identity (See Table 3).

**Table 3.** Regression Models for Health and Wellbeing (coefficients and Adj.$R^2$ shown are for final step with all variables included, $\Delta R^2$ indicates the change in $R^2$ at each step).

| Predictors: | Physical Health (Adj.$R^2$ = 0.098 ***) | | | Mental Health (Adj.$R^2$ = 0.135 ***) | | | Flourishing (Adj.$R^2$ = 0.231 ***) | | |
|---|---|---|---|---|---|---|---|---|---|
| | B | SE | B | B | SE | B | B | SE | B |
| **Step 1. Demographics** | | | | | | | | | |
| | $\Delta R^2$ = 0.063 *** | | | $\Delta R^2$ = 0.108 *** | | | $\Delta R^2$ = 0.056 *** | | |
| Sexual Identity | | | | | | | | | |
| Hetero/Straight | 0.118 | 0.091 | 0.061 | 0.378 | 0.097 | 0.179 *** | 1.62 | 0.565 | 0.124 ** |
| Gender Identity | | | | | | | | | |
| Transgender | −0.444 | 0.196 | −0.096 * | −0.272 | 0.207 | −0.055 | −1.97 | 1.21 | −0.064 |
| Non-binary | 0.072 | 0.144 | 0.022 | −0.147 | 0.152 | −0.042 | −0.427 | 0.886 | −0.020 |
| Cisgender male | 0.005 | 0.113 | 0.002 | 0.243 | 0.119 | 0.088 * | −0.340 | 0.709 | −0.020 |
| Racial Identity | | | | | | | | | |
| Another Race only Bi or Multiracial | −0.415 | 0.249 | −0.071 † | −0.153 | 0.262 | −0.024 | 0.068 | 1.53 | 0.002 |
| Latine only | 0.160 | 0.109 | 0.064 | 0.190 | 0.115 | 0.071 † | 0.075 | 0.673 | 0.004 |
| AAPI only | −0.611 | 0.176 | −0.149 *** | 0.004 | 0.186 | 0.001 | 1.05 | 1.09 | 0.039 |
| Black or African | −0.029 | 0.117 | −0.011 | 0.056 | 0.124 | 0.019 | −1.01 | 0.733 | −0.056 |
| American only | −0.279 | 0.175 | −0.068 | −0.177 | 0.184 | −0.040 | −0.560 | 1.08 | −0.021 |
| **Step 2. Activism Experience** | | | | | | | | | |
| | $\Delta R^2$ = 0.008 | | | $\Delta R^2$ = 0.001 | | | $\Delta R^2$ = 0.059 *** | | |
| Length of time | −0.023 | 0.037 | −0.030 | 0.016 | 0.039 | 0.020 | −0.032 | 0.226 | −0.006 |
| Effectiveness | 0.011 | 0.068 | 0.008 | −0.051 | 0.071 | −0.035 | 0.662 | 0.420 | 0.074 |
| **Step 3. Connectedness** | | | | | | | | | |
| | $\Delta R^2$ = 0.019 ** | | | $\Delta R^2$ = 0.010 * | | | $\Delta R^2$ = 0.072 *** | | |
| Belonging | 0.052 | 0.018 | 0.143 ** | 0.047 | 0.019 | 0.120 * | 0.553 | 0.111 | 0.229 *** |
| No. of Act. Groups/Orgs | 0.024 | 0.041 | 0.029 | 0.017 | 0.043 | 0.019 | 0.343 | 0.255 | 0.062 |
| **Step 4. Costs/Benefits** | | | | | | | | | |
| | $\Delta R^2$ = 0.033 *** | | | $\Delta R^2$ = 0.041 *** | | | $\Delta R^2$ = 0.066 *** | | |
| Costs | −0.014 | 0.003 | −0.211 *** | −0.017 | 0.003 | −0.228 *** | −0.114 | 0.020 | −0.248 *** |
| Benefits | 0.005 | 0.005 | 0.044 | −0.001 | 0.005 | −0.010 | 0.148 | 0.032 | 0.211 *** |

† $p < 0.10$, * $p < 0.05$, ** $p < 0.01$, *** $p < 0.001$. Note. The referent category for sexual identity is LGBQ; the referent category for gender identity is cisgender female; and the referent category for racial identity is white only.

### 6.1.1. Physical Health

The full model accounted for nearly 10% of the variance in physical health. Steps one, three, and four of the model added significant explanatory power to the variance in physical health, with step two (activism experience) not contributing. In the final fully specified model, youth identifying as transgender and youth identifying as Latine reported worse physical health outcomes (as compared to cisgender female youth and white youth, respectively). In addition, belonging to an activist community was positively related to

youth's physical health, while experiencing costs of activism was negatively related to physical health. There was no association between benefits of activism and physical health.

### 6.1.2. Mental Health

The full model accounted for 14% of the variance in mental health. All steps of the model added significant explanatory power to the variance in mental health, with the exception of step two, activism experience. In the final fully specified model, identifying as straight or heterosexual was related to better mental health than identifying as LGBQ. Neither the gender nor the race/ethnicity categories were predictive of mental health. Belonging to an activist community was positively related to youth's mental health, while experiencing costs of activism was negatively related to mental health.

### 6.1.3. Flourishing

The full model accounted for 23% of the variance in flourishing. Each step of the model added significant explanatory power to the variance in flourishing. In the final fully specified model, youth identifying as straight or heterosexual reported more flourishing than youth identifying as LGBQ. Neither the gender nor the race/ethnicity categories were predictive of flourishing. Belonging to an activist community and experiencing benefits of activism were positively related to flourishing, while experiencing costs of activism was negatively related to flourishing.

### 6.2. Qualitative Findings: "I Think Every Activist Has to Deal with Burnout"

Because the costs of activism were significantly associated with poorer wellbeing across all three regression models, we sought to explore costs further in the focus groups. Specifically, the most frequently endorsed costs were feeling guilty, tired, hopeless, and depressed because of one's activism, and because these costs are consistent with general understandings of the experience of burnout, we asked youth in the focus group to describe their experiences with burnout: what causes it, and what mitigates against it or sustains them in the face of it.

Across the focus groups, three main themes surfaced with regard to the sources of burnout: backlash; pressure to be the savior generation; and the lack of progress or change. In addition, as youth discussed the factors that helped to stave off or lessen burnout, peer support and belonging emerged as another central theme.

### 6.2.1. Backlash: Taking Heat and Hate

The youth activists in this study understood that being an activist requires expressing one's views and values in the face of those who disagree, and engaging with the naysayers can be both exhausting and self-injurious. As one respondent put it:

> Part of activism engagement is seeing what the other side, the oppressor, is doing and saying. That can get really, really, really hard. I think you can fall into a deep hole of reading people's hateful words towards you and your community, and that can get very toxic really quick.

Several respondents reported receiving responses to their activist-oriented social media posts that were "rude", "passive aggressive", or "hateful". Whether from strangers, family, or friends, these messages took a toll on activists, and contributed to burnout. As one LGBTQ activist explained:

> The issue with LGBTQ+ rights is that people other us. They're not seeing us as equals. And so to constantly be reminded that someone thinks of you as less than or as dirty . . . It doesn't matter if it's one-on-one, in person, or if it's online, [there are] things like that I don't like to debate, especially about my rights and things that I think are just human rights in general. Because I can't comprehend feeling the way they do. I don't know how to empathize with you if you don't

empathize with me. I don't know how to explain to you that you should care about my life. So, I face a lot of burnout from that.

Another activist who identified as Jewish and pro-Palestinian rights relayed an experience he had when a friend entered his "DMs and shared a lot of hateful rhetoric, under the guise of, it was my job to educate her, so she told me that I was a safe space". This painful and "suffocating" exchange inspired him, as it did for many others in the focus groups, to take a break from his activist work and set boundaries around time spent on social media.

### 6.2.2. Pressure to Be the Savior Generation

While contending with the hurtful rhetoric of those who belittle or disavow their activism engendered burnout for some youth activists, other youth activists identified just the opposite—adults' faith in them to "fix everything"—as the cause of their burnout. When asked about the root causes of the burnout she experienced, one activist recalled, "When I was a kid, I would often hear [messages] like, 'Your generation is going to fix the world. You're the solution to the problems'". Another echoed, "When we were growing up, I did hear that our generation could change the world". These messages from older activists and family members contributed to the pressure she felt. They told her, "It's basically going to go bad right after our generation, unless we completely stop this avalanche of injustice". Shouldering such heavy responsibility could sometimes feel overwhelming to youth. One college-aged activist recalled her own complicity in furthering this message of youth as saviors:

> I remember telling my sisters about this and then they felt like an intense wave of worry and anxiety about needing to fix everything. It's just a lot of pressure on younger kids too, especially when it's up to the adults who are in the position of power to fix things.

A male-identifying activist summed up the dilemma concisely: "It's definitely made me realize that we need to look out for the younger generation, but also it has caused a lot of burnout for me and I'm sure a lot of my peers". For some youth, the pressure not just to know how to solve the seemingly intractable problems adults have caused and passed on, but also to have the capacity to do so without adult assistance became disempowering. This pressure necessitated that they take a step back from the work to avoid completely burning out.

Even so, some activists acknowledged that it was empowering to hear these messages of adult confidence in their generation's capacity to effect change. As one indicated, "It was radical for someone to say that you can change the world". So, while these messages proved motivating for some youth, their repeated insistence added yet another layer of burden to others.

### 6.2.3. Lack of Progress and Change

A similar tension was at play for youth activists with respect to the lack of progress in the issues they cared about. On the one hand, youth felt discouraged when their efforts did not result in tangible or measurable change. They questioned whether their hard work had been worthwhile. On the other hand, they felt their engagement was all the more imperative to continue to chip away at such deeply entrenched problems. If they did not keep trying, then surely nothing would ever change. One respondent captured this duality:

> Never seeing positive results from what I'm doing ... Sometimes it can be demoralizing to not see any results when you've been doing something for years or [when] something has been happening for decades, and nothing has changed. That can be sort of demoralizing, and it makes you feel like, "Oh, this is sad. What am I really doing to help?" But then, also it can act as an activator almost, to make you want to feel more motivated to change it now.

Another respondent similarly felt that the lack of progress, while dispiriting and a potential source of burnout, was also a call to action. Reflecting on her experiences with burnout, she shared:

> The social justice issues that we're talking about are like so far gone, that we can kind of take steps forward, and we have to, but it's kind of slowly moving backwards, it seems ... We still have to try, even if we feel like it's not really accomplishing anything in the grand scheme of things.

Although their activism did not always result in the demonstrable social or institutional change that they hoped to accomplish, and this lack of progress was disappointing, even crushing for some, many of the youth activists in this study voiced a strong commitment to continue the work. Exploring the factors that sustained them in these efforts, despite the risks of burnout, is where we turn next.

### 6.2.4. Belonging to an Activist Community and Peer Support as Salves for Burnout

Several of the youth activists in the focus groups described burnout as inevitable, an inescapable aspect of their work as activists. As a result, they stressed the importance of developing strategies to contend with it. As one respondent said, "Managing burnout ... is really important in the long term because we're trying to be like lifetime activists, and it's not going to get any easier". These strategies ranged from limiting time spent on social media to taking intentional breaks from the work to pursuing self-care routines; however, the most commonly discussed strategy involved leaning on peers in the activist community for support and camaraderie.

Feeling like they were not alone and that they had a community that had their backs, understood their struggles, and would support them when they were discouraged or down helped activists remain engaged when burnout began to set in. One respondent shared, "It's always helpful to surround myself with like-minded people who care about the same issues, who are fighting in the same fight with me, because that way you can like really empathize with one another when you're facing burnout". Another respondent echoed:

> I'd like to add to [the discussion of] facing burnout and combating that. One of the ways I do that is conversing with people who are involved in the fight or in activism too, and that kind of re-motivates me and lets me know that I'm not alone in what I'm doing.

In addition to describing their activist community as a space where they can "re-energize ourselves", respondents described it as an "affirming" and "comforting" space of "true support and understanding". As one put it, "For me, combating burnout was through turning inwards and finding peace and support in my [fellow activist] communities". Another respondent explained how she derived inspiration and ongoing motivation from the activist community:

> I think just seeing people who care so deeply about the same issues that you also care about is just so heartening, so I think that's what really keeps me loving activism: just seeing other people who are so willing to make a change for such an important issue.

For many of the youth in this study, belonging to an activist community generated profound feelings of support and sustenance as they pursued the long, hard work of activism.

## 7. Discussion

This study examined the costs and benefits of youth activism and explored how these costs and benefits relate to youth activists' reports of physical health, mental health, and flourishing. Survey data from a diverse national sample of young activists revealed that overall, youth report more benefits from their activism than costs; however, costs are significantly associated with poorer health outcomes across all three outcome variables.

The more costs youth indicate experiencing, the worse physical health, worse mental health, and lower levels of flourishing they report. Benefits were positively associated with flourishing only.

Focus group data provided some insight into the ways in which the costs of activism take a negative toll on youth, predominantly through burnout. Understood as a constellation of emotions, including fatigue, hopelessness, and depression that set in after a period of sustained and marked exertion, burnout is a commonly reported cost of activism for this sample. The focus groups shed light on three main sources of burnout: facing backlash from those who disagree with their efforts; feeling generational pressure to solve the world's most pressing problems; and facing a lack of progress or change. As draining and dispiriting as these root causes of burnout can be, belonging to an activist community can help soften their ill-effects.

Analysis of the survey data found that a sense of belonging to an activist community was statistically associated with all three health outcomes. The stronger the sense of belonging the youth activist reported, the better their mental health, physical health, and level of flourishing. Consistent with Ballard and Ozer's [9] speculation that social capital and connection to others is a pathway linking activism to wellbeing, other work has shown how a sense of belonging can serve as a powerful protective factor for youth activists' health and wellbeing, even in the face of the deleterious effects of the costs of activism [5,14,66].

A sense of belonging to an activist community also emerged as a prominent theme in focus group conversations about the factors that sustain youth activists when they experience burnout. Focus group respondents turned to their peers within the activist community for support, empathy, inspiration, and motivation when feelings associated with burnout took hold. Many credited their membership in the activist community with keeping them engaged in the struggle.

This mixed-methods study contributes to the body of knowledge about the impacts of activism on youth in several ways. First, it shows that LGBQ youth activists report significantly worse mental health and lower flourishing scores than their straight counterparts, and that their sexual identities are significantly associated with these two dimensions of wellbeing, even when holding costs and benefits constant. This finding extends the extant literature on mental wellbeing disparities among the general population of LGBQ youth [67,68] to LGBQ youth activists. Interestingly, LGBQ youth reported deriving greater benefits from their activism than straight youth, though they also reported experiencing greater costs. Similarly, identifying as transgender was associated with worse physical health outcomes. While transgender youth reported experiencing greater benefits from their activism than cisgender males, they also reported more costs than either cisgender females or cisgender males. In focus groups, LGBTQ youth explained how exhausting and dehumanizing it can be to constantly have to combat homophobia or transphobia and defend their right to exist. Other work has found that among LGBTQ youth, activism may serve a protective function, mediating the relationships between economic precarity and health problems and between minority stress and health problems [15]. Activism has also been associated with less psychological distress and suicidal ideation among LGBTQ youth, even when controlling for effects of bullying and discrimination [69]. However, some research suggests that LGBTQ-specific activism may play a different role than activism unrelated to one's own identity, as one study found no direct relationship between participation in LGBTQ-specific activism and depression among LGBTQ college students, even when controlling for experiences of sexual-orientation victimization and discrimination [16]. Our study adds to this literature by showing that although activism may help lessen some of the distress LGBTQ youth may experience, it is not without costs and adverse health effects. Youth activists and those who support them should be particularly attentive to the unique stressors faced by LGBTQ youth activists, as these costs do impinge on their wellbeing. It is possible that the healing justice framework that has been used so profitably in settings with Black and other racially minoritized youth activists (see [3,4]) could be adapted for youth activists facing oppression due to their gender or sexual identities.

This study also adds to the literature on youth activism and wellbeing by including youth of different ages and examining differences according to their educational profiles. Given the significant associations we find between costs of activism and wellbeing, the finding that college students incur significantly more costs related to their activism than high school students may help explain why so much of the research on adolescent activist populations turns up positive implications for health, while the research on college-aged activists populations is more mixed. Although the two groups did not differ in their reported benefits, benefits were only associated with flourishing. It is nonetheless noteworthy that youth activists who were not enrolled in school reported fewer benefits from their activism than college students. While being a member of campus or school-based organizations or clubs may facilitate benefits, such as friendships and belonging, for youth enrolled as students, whether and how non-enrolled youth find such community is an open question; however, it is clear that this is a population of youth that has been largely neglected in extant research on youth activism and one worthy of further study.

The lack of significant differences in mental health and flourishing by racial and ethnic identities is surprising, given existing literature, which has found that youth of color, particularly Black and Latine youth, tend to have poorer health outcomes than white youth [70,71]. Our lack of significant racial/ethnic differences may be due to our sample, as we specifically studied youth activists as opposed to general populations of youth. Indeed, one study of college LGBTQ student activists suggests that racial differences in mental health outcomes may not exist, at least among some identity groups of youth activists [16]. The lack of significant racial/ethnic differences in the costs and benefits of activism by race/ethnicity found in this study likewise does not align with much of the prior research, which suggests activism can bring particular benefits [22,48,51] and particular costs [16,57] for non-white youth. Our non-significant findings may have been a function of using a summative measure; statistical differences might emerge when specific costs or specific benefits within the larger scale are examined. Furthermore, group-specific measures of the costs and benefits of activism could be explored to make sure unique costs and benefits for specific racial and ethnic groups are captured, which may also provide more insight into the consequences of activism for health.

In addition to adding to our understanding of how various dimensions of identity shape health outcomes for youth activists, this study highlights the salience of belonging both to promoting positive health and as an antidote to burnout. Building on other research that shows how a sense of connectedness to a collective and social capital can serve as a protective mental health pathway for youth activists [5,9,14,66], this finding underscores the value of feeling a sense of membership in an activist community. Interestingly, the number of organizations or groups to which one belonged did not matter as much as feeling connected to any activist community. This finding gives youth activists permission to scale down their membership efforts and invest time and care in nurturing one community or the broader activist community, rather than spreading themselves thinly across several organizations. It may also speak to the significance of programs like the ACLU Advocacy Institute that bring together youth activists from across geographies, issue areas, and backgrounds to build relationships and feel connected to a wider network. The development of camaraderie among marginalized youth, in particular, may enhance marginalized youth's understanding of systemic inequalities and increase their engagement in activism to fight against shared oppression [43].

Finally, this study illuminates the complex relationship between costs and benefits of activism as they shape youth wellbeing. While scholars are starting to acknowledge that both positive and ill effects accrue from participating in youth activism [4,5,9], few have sought to compare them quantitatively or test them in relation to mental health and physical health. Although youth activists in this study report experiencing more benefits than costs, the costs are more consistently linked to worse wellbeing outcomes, with benefits showing a significant relationship only to flourishing. Therefore, efforts to minimize costs while

heightening a sense of belonging, which was not among the benefits measured in this study, appear warranted.

## 8. Limitations and Future Directions

The study's findings are limited to self-reports from a self-selected sample of youth who chose to participate in the ACLU Advocacy Institute and then volunteered to take part in the study. While the sample was diverse in some regards, it was largely composed of youth who are white, middle and upper-middle class, high school aged, LGBQ, cisgender, and female. In this study, we examined these identity categories as discrete variables, and we did not explore intersectional identities. We also did not include disability and socioeconomic status in our analyses. Future research could build on this study by using an intersectional lens to examine the costs-benefits calculus and health implications for specific groups of youth, such as queer Black girls or low-income transgender youth. In addition, rather than using dimensions of identity as control variables, future researchers could explore whether and how youths' various identities might moderate the relationship between their activism and their wellbeing. Such analysis is especially important with regard to race and ethnicity, as our study was limited both because of the relatively smaller numbers of participants in the various non-white racial and ethnic groups and because we used "white" as the referent category in our models, thereby limiting our understanding of how health outcomes compare among non-white youth once activism and its attendant costs and benefits are accounted for.

Another limitation of the study was its reliance on single item measures for physical health, mental health, and perceived effectiveness of activism; however, single items may be appropriate in some cases. Drawing on psychological research, Bowling [63] argues "the single item question can provide valuable information, has the advantage of simplicity, and can be reliable and valid" (p. 343). Future research could test the relationships examined in this study using more extensive, previously validated measures of mental and physical health.

This study's cross-sectional design precludes our understanding of directionality in the associations we examined. It could be that health and wellbeing could influence the extent to which youth can be involved as activists or feel like they belong in activist spaces. Future longitudinal research could explore these relationships over time. Researchers might also examine whether and how the specific issue areas in which activists are working shape their experiences of costs and benefits and their wellbeing. Finally, researchers can use factor analyses to identify the different types of costs and benefits youth activists report to see if any exert a differential impact with regard to wellbeing for particular youth.

Activism is a complex undertaking, with variable costs and benefits. Although this study cannot answer all questions about how such costs and benefits affect the wellbeing of youth with different intersectional identities, it does bring us a step closer to understanding the prevalence and sources of burnout among youth activists as well as the key role a sense of belonging to an activist community plays in helping them to sustain their efforts and maintain their health.

**Author Contributions:** Conceptualization, J.O.C., E.G., C.D.E. and L.W.-L.; methodology, E.G. and C.D.E.; formal analysis, J.O.C. and C.D.E.; investigation, E.G. and C.D.E.; data curation, E.G. and C.D.E.; writing—original draft, J.O.C.; writing—review and editing, J.O.C., E.G., C.D.E. and L.W.-L.; project administration, E.G. All authors have read and agreed to the published version of the manuscript.

**Funding:** This research received no external funding.

**Institutional Review Board Statement:** The study was conducted in accordance with the Declaration of Helsinki, and approved by the Institutional Review Board (or Ethics Committee) of UCLA (protocol code 22-001642).

**Informed Consent Statement:** Informed consent or assent was obtained from all subjects involved in the study.

**Data Availability Statement:** The data are not publicly available because participants have not provided permission for their data to be shared.

**Acknowledgments:** The authors express their gratitude to the ACLU National Advocacy Staff, for their support with data collection, input on survey development, and feedback on this manuscript; to Mya Haynes, former ACLU Research Intern, for her work on initial survey development; to former ACLU Research Intern Marcos Viveros-Cespedes and former ACLU Research Fellow Cyrus O'Brien for participating in focus group facilitation; and to the ACLU Youth Activism Research Collaborative's Youth Advisory Board for their thoughtful review and interpretation of our findings.

**Conflicts of Interest:** The authors declare no conflict of interest.

## Appendix A Costs and Benefits of Activism Scales

Answer choices on a seven-point Likert scale range from strongly disagree to strongly agree. The Political Costs of Activism (adapted from Smith et al. [61])

1. I have felt tired as a result of my activism.
2. I have felt depressed as a result of my activism.
3. I have lost sleep because of my activism.
4. I have felt hopeless as a result of my activism.
5. I have felt guilty for not doing enough in my activism.
6. My activism has created problems for me with my family.
7. My activism has created problems for me with my peers.
8. My activism has created problems for me with academics/school work.
9. My activism has created problems for me with the police.
10. I have experienced harassment due to my activism.
11. I have experienced identity-based discrimination due to my activism.
12. I have had traumatic experiences due to my activism.

The Political Benefits of Activism (adapted from Oosterhoff et al. [62])

1. My political and social views are an important part of who I am.
2. On occasion, I have been proud of comments I made during a political discussion.
3. I have posted or written political things online that I am proud of.
4. Activism has given me a chance to demonstrate good judgment.
5. Activism has contributed to my personal growth.
6. Engaging in activism allows me to express myself.
7. Engaging in activism makes me fulfilled.
8. Being an activist has given me a sense of purpose.
9. Being an activist has made me feel empowered.
10. Similarities in political and social views have brought my extended family closer together.
11. On occasion, activism has made my home life more pleasant.
12. Similarities in political and social views have strengthened a friendship I valued.

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
