# Peer review of "Burnout and Belonging: How the Costs and Benefits of Youth Activism Affect Youth Health and Wellbeing"

_2673-995X, doi:10.3390/youth3010009_

Round 1

Reviewer 1 Report

It was a pleasure to review this manuscript. The study is well-designed and very nicely based in theory and existing evidence. The set of findings are important, theoretically-consistent, and present novel insights that I was excited to read and I think will be relevant and useful to the field. I commend the authors on very clear writing and structure as well as on employing a Youth Advisory Committee to shape the study and provide insights about results.  I hope the authors find these comments helpful.

·         Regarding the study design, were the quantitative and qualitative components conducted in parallel, or were the focus groups designed to understand the survey findings (as is implied in the first sentence of “qualitative findings” section).

·         In the results section, the demographic findings are presented as “in a final step” of the regression but from the tables, it looks like they were entered in Step 1. Please clarify this. In addition, the demographic findings reported are split across 2 places in the Mental Health and Flourishing findings paragraph.

·         It would be helpful to have more context about the sample participating in the ACLU program; specifically, this is a group of people who identify as activists, which is a high bar. Many young people engage in activist work but do not necessarily identify as “An Activists.” How was the program advertised, who applied, and might the sample affect generalizability of results (beyond those noted in the limitations section)?

·         The discussion of findings for LGBQ youth in the Discussion is interesting; I would like to hear more about what the authors make of (mostly NS) findings about both costs/benefits of activism and WB outcomes across race/ethnicity (given prior literature).

·         There were some limitations I noted as I read (e.g., measures, cross-sectional) but they were appropriately addressed in the limitations section.

Author Response

Dear Reviewer 1,

Thank you for your support of our manuscript. We appreciated your constructive suggestions and indeed found them quite helpful. We have made the following changes in response to your feedback:

  • We have clarified that the survey preceded the focus groups, and that the focus groups were used in part to explore interesting findings that emerged from analysis of the survey data.
  • We have edited the text of both our analytical approach and our findings to clarify that the demographic variables were entered in Step 1 of the regression models. Thank you for pointing out the confusion that arose from our split presentation.
  • We added more context about the sample at the outset of the methods section. We described how the program is advertised and what the application process looks like. Incidentally, we agree with you that some young people do not identify as activists, even though they are engaged in activist work. This was a topic that emerged in some of our focus group discussions, and we have added a question about this self-identification to our survey for future administration. Although we refer to our sample as youth activists in this study, we have added a footnote to reflect your point that not all participants necessarily identify as such.
  • We have added reflections on the non-significant findings in the health and the costs/benefits of activism across race and ethnicity in the discussion. Here, we note that we were surprised by these findings, given the extant literature, and we speculate that the results may have been a function of using summative measures, rather than fine-grained or group-specific measures. Also, in the limitations section, we have added text to acknowledge the small sample sizes in the non-white racial/ethnic groups and the need for future research to use demographic variables as moderators rather than covariates, when seeking to understand the relationship between activism and health for youth of different intersectional identities.

In addition to these changes, we have incorporated additional references that Reviewer 2 recommended. We have also edited the entire manuscript, polishing the language. We believe that addressing your feedback has helped us improve the manuscript markedly, and we extend our deep appreciation for the careful attention you paid to it.  

Reviewer 2 Report

Excellent scholarship- well designed, justified, explained, and contextualized. I have a couple suggestions for scholarship that would also contribute to how the findings were contextualized:

Anyiwo, N., Palmer, G. J., Garrett, J. M., Starck, J. G., & Hope, E. C. (2020). Racial and political resistance: An examination of the sociopolitical action of racially marginalized youth. Current opinion in psychology35, 86-91.

Anyiwo, N., Bañales, J., Rowley, S. J., Watkins, D. C., & Richards‐Schuster, K. (2018). Sociocultural influences on the sociopolitical development of African American youth. Child Development Perspectives12(3), 165-170.

Finlay, A., Wray‐Lake, L., & Flanagan, C. (2010). Civic engagement during the transition to adulthood: Developmental opportunities and social policies at a critical juncture. Handbook of research on civic engagement in youth, 277-305.

Kulick, A., Wernick, L. J., Woodford, M. R., & Renn, K. (2017). Heterosexism, depression, and campus engagement among LGBTQ college students: Intersectional differences and opportunities for healing. Journal of homosexuality64(8), 1125-1141.

Wernick, L. J., Kulick, A., & Woodford, M. R. (2014). How theater within a transformative organizing framework cultivates individual and collective empowerment among LGBTQQ youth. Journal of Community Psychology42(7), 838-853.

Author Response

Dear Reviewer 2,

Thank you for your kind words about our manuscript and your support of the work. We appreciated the five additional references you recommended to us, and we have integrated citations to each of these relevant works throughout the revised manuscript.

In addition to these additions, we have made a few changes in response to feedback from Reviewer 1, including clarifying our analytic approach, adding more details about our sample, and discussing the non-significant differences in health outcomes and costs/benefits by race/ethnicity. We have also edited the entire manuscript, polishing the language. We believe that addressing your feedback has helped us improve the manuscript, and we extend our appreciation to you.